
# Correcting Thornthwaite potential evapotranspiration using a global grid of local coefficients to support temperature-based estimations of reference evapotranspiration and aridity indices

Vassilis Aschonitis[1], Dimos Touloumidis[2], Marie-Claire ten Veldhuis[2], Miriam Coenders-Gerrits[2]

[1]Soil and Water Resources Institute, Hellenic Agricultural Organization – DEMETER, Thessaloniki – Thermi, 57001, Greece
[2]Water Resources Section, Delft University of Technology, Stevinweg 1, 2628 CN Delft, the Netherlands

*Correspondence to*: Vassilis Aschonitis (v.aschonitis@swri.gr)

**Abstract.** Thornthwaite's formula is globally an optimum candidate for large scale applications of potential evapotranspiration
and aridity assessment at different climates and landscapes since it has the lower data requirements compared to other methods
and especially from the ASCE-standardized reference evapotranspiration (former FAO-56), which is the most data demanding
method and is commonly used as benchmark method. The aim of the study is to develop a global database of local coefficients
for correcting the formula of monthly Thornthwaite potential evapotranspiration ($E_p$) using as benchmark the ASCE-
standardized reference evapotranspiration method ($E_r$). The validity of the database will be verified by testing the hypothesis
that a local correction coefficient, which integrates the local mean effect of wind speed, humidity and solar radiation, can
improve the performance of the original Thornthwaite formula. The database of local correction coefficients was developed
using global gridded temperature and $E_r$ data of the period 1950-2000 at 30 arc-sec resolution (~1 km at equator) from freely
available climate geodatabases. The correction coefficients were produced as partial weighted averages of monthly $E_r/E_p$ ratios
by setting the ratios' weight according to the monthly $E_r$ magnitude and by excluding colder months with monthly values of
$E_r$ or $E_p$ <45 mm month$^{-1}$ because their ratio becomes highly unstable for low temperatures. The validation of the correction
coefficients was made using raw data from 525 stations of Europe, California-USA and Australia including data up to 2020.
The validation procedure showed that the corrected Thornthwaite formula $E_{ps}$ using local coefficients led to a reduction of
RMSE from 37.2  to 30.0 mm m$^{-1}$ for monthly and from 388.8 to 174.8 mm y$^{-1}$ for annual step estimations compared to $E_p$
using as benchmark the values of $E_r$ method. The corrected $E_{ps}$ and the original $E_p$ Thornthwaite formulas were also evaluated
by their use in Thornthwaite and UNEP (United Nations Environment Program) aridity indices using as benchmark the
respective indices estimated by $E_r$. The analysis was made using the validation data of the stations and the results showed that
the correction of Thornthwaite formula using local coefficients increased the accuracy of detecting identical aridity classes
with $E_r$ from 63% to 76% for the case of Thornthwaite classification, and from 76% to 93%  for the case of UNEP classification.
The performance of both aridity indices using the corrected formula was extremely improved in the case of non-humid classes.
The global database of local correction factors can support applications of reference evapotranspiration and aridity indices
assessment with the minimum data requirements (i.e. temperature) for locations where climatic data are limited. The global



grids of local correction coefficients for Thornthwaite formula produced in this study are archived in PANGAEA database and can be assessed using the following link: https://doi.pangaea.de/10.1594/PANGAEA.932638 (Aschonitis et al., 2021).

## 1 Introduction

The assessment of potential or reference evapotranspiration is among the most important components for many hydro-climatic
applications such as irrigation design and management, water balance assessment studies, and assessment of aridity classification and drought indices (Weiß and Menzel, 2008; Wang and Dickinson, 2012; McHanon, 2013; Aschonitis et al., 2017).

Such applications, and especially applications of aridity classification and drought indices (UNEP 1997; Thornthwaite, 1948; Palmer, 1965; Holdridge, 1967; Beguería et al., 2014) that are usually employed at large scales, require estimations of
potential or reference evapotranspiration of respective scale. The major problem in such applications is not only the limited availability of stations per se but also the limitation of many stations to provide data for a complete set of parameters (i.e. precipitation, temperature, solar radiation, wind speed, humidity). A complete set of climate parameters is prerequisite for accurate estimations of potential or reference evapotranspiration using integrated methods such as these of Penman (1948), Shuttleworth, 1993; Allen et al. (1998), Allen et al. (2005), and others which are expressions of energy balance. Unfortunately,
large scale applications suffer from these limitations and the common solution is to use temperature-based formulas (Thornthwaite, 1948; McCloud, 1955; Hamon 1961, 1963; Baier and Robertson, 1965; Malmström, 1969; Hargreaves et al., 1982; Camargo et al., 1999; Droogers and Allen 2002; Pereira and Pruitt 2004; Oudin et al., 2004; Trajkovic 2005, 2007; Trajkovic and Kolakovic, 2009a,b; Almorox et al., 2015; Aschonitis et al. 2017; Sanikhani et al. 2019; Quej et al., 2019; Trajkovic et al., 2020). However, extensive literature shows that temperature-based formulas are inherently of low performance
because temperature cannot describe properly the evaporative flux, while various studies have shown differences among the Penman–Monteith-based and temperature-based potential evapotranspiration assessments such as the one of Thornthwaite (1948), which is the most popular in aridity and drought indices applications (Sheffield et al., 2012; Dai, 2013; van der Schrier et al., 2013; Trenberth et al., 2014; Yuan and Quiring, 2014; Zhang et al., 2015; Asadi Zarch et al., 2015).

The formula of Thornthwaite (1948) was firstly proposed as internal part of the respective Thornthwaite
aridity/humidity index and it was calibrated based on measured monthly evapotranspiration from some well-watered grass-covered lysimeters in the eastern and central USA (Willmott et al.1985; Van Der Schrier et al., 2011). The specific formula overestimates the potential evapotranspiration in humid climates and underestimates it in arid climates (Pereira and Pruitt, 2004; Castaneda and Rao, 2005; Trajkovic and Kolakovic, 2009), and thus, a number of efforts have been made to amend the parameters or constants of the empirical formula to adapt it to various geographical zones (Jain and Sinai, 1985; Pereira and
Pruitt, 2004; Castaneda and Rao, 2005; Zhang et al., 2008; Bakundukize et al., 2011; Yang et al., 2017). Indicative modifications were proposed by Willmott et al. (1985) using an additional parametrization presented for mean monthly temperature above 26.5 ℃ and an adjustment for variable daylight and month lengths. Camargo et al. (1999) substituted the





mean monthly temperature by another factor called effective temperature considering the amplitude between maximum and
minimum temperature. Jain and Sinai (1985) modified the constant in the general formula based on the min-max range of the
annual mean air temperature to calculate the evapotranspiration for semiarid conditions. Pereira and Pruitt (2004) proposed an
adaptation of the Thornthwaite scheme to estimate the daily reference evapotranspiration on two contrasting environments in
USA and Brazil. Castaneda and Rao (2005) recalibrated the coefficient of the general formula based on estimations of potential
evapotranspiration using the FAO Penman-Monteith method in southern California. Zhang et al. (2008) used a modified
formula to estimate the actual evapotranspiration in cropland, shrubland and forest located in the subalpine region of
southwestern China. Bakundukize et al. (2011) used two modifications and the original Thornthwaite method to groundwater
recharge estimations in the inter-lacustrine zone of East Africa. Yang et al. (2017) presented a method to quantitatively identify
the differences in the spatiotemporal variabilities of global drylands between the Thornthwaite and Penman–Monteith
parameterizations.

The last years, advanced interpolation techniques, climatic models and other methods have achieved to generate gridded
datasets of various climatic parameters (Hijmans et al., 2005; Sheffield et al., 2006; Osborn and Jones, 2014; Harris et al.,
2014; Brinckmann et al., 2016; Liu et al. 2020) facilitating attempts to develop global maps of potential/reference
evapotranspiration and to investigate the accuracy of formulas of reduced parameters versus benchmark methods  at global
scale (Droogers and Allen, 2002; Weiß and Menzel, 2008; Zomer et al., 2008; Aschonitis et al., 2017). A similar attempt is
performed in this study aiming to develop a global database of local correction coefficients for the original Thornthwaite
formula that will better support all hydro-climatic applications and especially to support large scale applications of aridity
indices, which are highly prone to data limitations. The hypothesis that is tested in this work is that a local correction coefficient
that integrates the local mean effect of wind speed, humidity and solar radiation can improve the performance of the original
Thornthwaite formula and to convert it at the same time to a formula of reference evapotranspiration for short reference crop.

## 2 Data and Methods

### 2.1 Data

The methodological steps of the next sections are used to develop a global map of local coefficients for correcting the original
potential evapotranspiration formula of Thornthwaite following a calibration and a validation procedure.

The derivation/calibration procedure was performed at global scale using global gridded data from two databases. The
first database of Hijmans et al. (2005) provides gridded data of mean monthly precipitation $P$ and mean monthly temperature
$T$ of the period 1950-2000 (WorldClim version 1.2) at 30 arc-sec spatial resolution (~1×1 km at the equator) (Fig.1a,b). The
second database is of Aschonitis et al. (2017), which provides gridded data of mean monthly reference evapotranspiration $E_r$
of the period 1950-2000 at five different resolutions (30 arc-sec, 2.5 arc-min, 5 arc-min, 10 arc-min and 0.5 deg) (Fig.1c). The
method used for estimating $E_r$ is the ASCE-standardized method (former FAO-56), which estimates reference
evapotranspiration for short clipped grass (Allen et al., 2005). The database of $E_r$ (Aschonitis et al., 2017) was built using the





temperature from the first database of Hijmans et al. (2005) at 30 arc-sec resolution and for this reason the two gridded

databases are compatible.

**[FIGURE 1]**

The validation procedure was performed using raw data of stations from three different databases. The first database is

the CIMIS database (California Irrigation Management System – CIMIS, http://www.cimis.water.ca.gov), which includes

stations from California-USA and it was selected because it provides a dense and descriptive network of stations for a specific

region that combines semi-arid/temperate coastal, plain, mountain environments. In total 60 stations (Fig.2a) were used from

CIMIS database that have at least 15 years of observations with a significant part of their observations after 2000. The second

database is the AGBM database (Australian Government – Bureau of Meteorology, http://www.bom.gov.au). This database

includes many stations from Australia and was selected because the station's network covers a large territory with a large

variety of climate classes from desert to tropical climate. The selection of stations was performed in order to cover all the

possible existing Köppen-Geiger climatic types (Peel et al., 2007) and altitude ranges that exist in the Australian territory. In

total 80 stations were used (Fig.2b), that have at least 15 years of observations with a significant part of their observations after

2000. The third database is the ECAD database (European Climate Assessment & Database, https://www.ecad.eu). This

database is a network that contains more than 20,000 stations throughout Europe and provides daily observations of

climatological parameters. In this study, a final number of 385 stations (Fig.2c) was selected because they contained complete

data of precipitation, temperature, solar radiation, relative humidity and wind speed for a period of at least 20 years with a

significant part of their observations after 2000. Some additional stations from the three databases (CIMIS, AGBM, ECAD),

which do not have at least 15 years of observations, were selected due to their special climate Köppen-Geiger class or the high

altitude of their location). The total number of stations used in the study from the three databases is 525 and their full description

is given in Table S1 of the Supplementary material.

**[FIGURE 2]**

**2.2 Derivation and validation of Thornthwaite correction coefficients for short reference crop based on ASCE-**
**standardized method**

The monthly potential evapotranspiration $E_p$ using the Thornthwaite (1948) method after its adjustment for variable daylight

and month lengths (Willmott et al., 1985) is estimated as follows:

$$E_p = 16 \cdot \left(\frac{10 \cdot T_{mean}}{J}\right)^a \cdot \frac{N \cdot n}{365} \tag{1}$$

$$J = \sum_{i=1}^{12} j_i \tag{2a,b,c}$$





$$j_i = \left(\frac{T_{mean,\,i}}{5}\right)^{1.514}$$

$$\alpha = (6.75 \cdot 10^{-7}) \cdot J^3 - (7.71 \cdot 10^{-5}) \cdot J^2 + (1.79 \cdot 10^{-2}) \cdot J + 0.492$$

$$N = \frac{24}{\pi} \cdot \omega_s$$

$$\omega_s = \frac{\pi}{2} - \arctan\left[\frac{-tan(\varphi) \cdot tan(\delta)}{X^{0.5}}\right]$$

where: $X = 1 - [\tan(\varphi)]^2 \cdot [\tan(\delta)]^2$, if $X \leq 0$ then $X = 0.00001$

$$\delta = 0.409 \cdot sin\left(2 \cdot \pi \cdot \frac{d_j}{365} - 1.39\right)$$

(3a,b,c,d)

where $E_p$: the mean monthly potential evapotranspiration or potential evapotranspiration of month $i$ (mm month$^{-1}$), $T_{mean,i}$: the mean monthly temperature (ºC), $n$: the number of days in the month, $N$: the mean length of daylight of the days of the month (hours), $J$: the annual heat index, $j_i$: the monthly heat index, $\alpha$: the function of the annual heat index and $d_j$: the Julian day.

The benchmark method that was used for developing correction coefficients for the temperature-based method of Thornthwaite $E_p$ is ASCE-standardized method (former FAO-56), which estimates reference evapotranspiration from short clipped grass, as follows (Allen, et al., 2005):

$$E_r = \frac{0.408 \cdot \Delta \cdot (R_n - G) + \frac{\gamma \cdot u_2 \cdot (e_s - e_a) \cdot C_n}{(T_{\text{mean}} + 273.16)}}{\Delta + \gamma \cdot (1 + C_d \cdot u_2)}$$

(4)

where $E_r$: the reference evapotranspiration (mm d$^{-1}$), $\Delta$: the slope of the saturation vapour pressure-temperature curve (kPa ºC$^{-1}$), $R_n$: the net radiation at the crop surface (MJ m$^{-2}$ d$^{-1}$), $G$: the soil heat flux density at the soil surface (MJ m$^{-2}$ d$^{-1}$), $\gamma$: the psychrometric constant (kPa ºC$^{-1}$), $u_2$: the wind speed at 2 m height above the soil surface (m s$^{-1}$), $e_s$: the saturation vapour pressure (kPa), $e_a$: the actual vapour pressure (kPa), $T_{\text{mean}}$: the mean daily air temperature (ºC), $C_n$ and $C_d$: constants, which vary according to the time step and the reference crop type and describe the bulk surface resistance and aerodynamic roughness. Eq.3 can be applied for two types of reference crop (i.e. short and tall). The short reference crop (ASCE-short) corresponds to clipped grass of 12 cm height and surface resistance of 70 s m$^{-1}$ where the constants $C_n$ and $C_d$ have the values 900 and 0.34, respectively. (Allen et al., 2005). The use of Eq.3 in daily or monthly step for short reference crop is equivalent to FAO-56 method (Allen et al., 1998) and this is how it is used in this study.

The derivation of a correction coefficient for Eq.1 using as benchmark the values of Eq.3 is performed based on the same procedure proposed by Aschonitis et al. (2017) that has been used before for developing partial weighted annual correction coefficients for Priestley-Taylor and Hargreaves-Samani evapotranspiration methods. The procedure starts with the derivation of the monthly coefficient $c_{th,i}$ for each month $i$ based on Eq.5. Applying this procedure, twelve values of monthly $c_{th,i}$ are produced. The 12 monthly $c_{th}$ coefficients are then used to build mean annual coefficients. As it was mentioned in Aschonitis et al. (2017), the efficiency of mean annual correction coefficients is mainly associated to their ability to better



describe the larger values of the dependent variable (i.e. the values of $E_r$ during summer/hot months) and not the smaller values

during cold period where the absolute errors ($e_i = E_{r,i} - E_{p,i}$) are smaller. For this reason, weighted annual averages based on the monthly $c_{th,i}$ coefficients are estimated considering the participation weight of each month in the annual $E_r$. Moreover, under cold conditions, the monthly coefficients $c_{th,i}$ may present unrealistic values that significantly affect the weighted averages. To solve this problem, threshold values for the monthly $E_{p,i}$ and $E_{r,i}$ were used before the inclusion of their $c_{th,i}$ in the weighted average estimations. Preliminary analysis showed that when the mean monthly $E_{p,i}$ and/or $E_{r,i}$ are below ~45 mm month$^{-1}$ (~1.5

mm d$^{-1}$), then unrealistic mean monthly $c_{th,i}$ values occur (as unrealistic values are considered those, which are at least one order of magnitude larger or smaller from 1). Taking into account the above, the following procedure was performed in order to obtain a partial weighted average based on monthly $c_{th,i}$ values after excluding those months with $E_r$ and/or $E_p \leq 45$ mm month$^{-1}$ as follows:

$$c_{th,i} = E_{r,i}/E_{p,i} \tag{5}$$

$$\text{If } E_{r,i} > 45 \text{ mm month}^{-1} \text{ then } F_{r,i} = 1 \text{ else } = 0 \tag{6}$$

$$\text{If } E_{p,i} > 45 \text{ mm month}^{-1} \text{ then } F_{m,i} = 1 \text{ else } = 0 \tag{7}$$

$$E_{r,i}^{adj} = E_{r,i} \cdot F_{r,i} \cdot F_{m,i} \tag{8}$$

$$AE_r^{adj} = \sum_{i=1}^{12} \left( E_{r,i}^{adj} \right) \tag{9}$$

$$C_{th} = \sum_{i=1}^{12} \left( \frac{E_{r,i}^{adj}}{AE_r^{adj}} \cdot c_{th,i} \right) \tag{10}$$

where $c_{th,i}$: the monthly correction coefficient, $F_{r,i}$: the filter function for the reference method (ASCE) with values 0 or 1, $F_{m,i}$:

the filter function for the understudy model (Thornthwaite formula) with values 0 or 1, $E_{r,i}^{adj}$: the adjusted monthly value of $E_{r,i}$ from ASCE-short method that becomes 0 when $F_{r,i}$ or $F_{m,i}$ is 0, $AE_r^{adj}$: the annual sum of the monthly $E_{r,i}^{adj}$ adjusted values, $C_{th}$: the annual partial weighted average (p.w.a.) of the monthly $c_{th,i}$ coefficients for short reference crop and $i$: the index of each month. Considering the above, the final corrected Thornthwaite formula for monthly calculations is given by the following equation:

$$E_{ps,i} = C_{th} \cdot E_{p,i} \tag{11}$$

where $E_{ps,i}$: the corrected temperature-based short reference crop evapotranspiration (mm month$^{-1}$) of month $i$.

         The above procedure was followed in order to calibrate the annual partial weighted average $C_{th}$ (Eq.10) for every location on the globe based on mean monthly $E_r$ and $E_p$ of 1950-2000 using:

• the gridded mean monthly temperature data of Hijmans et al. (2005) that were further used to estimate the mean monthly gridded original Thornthwaite $E_p$ (Eq.1) for the period 1950-2000 (in the form of 12 raster datasets of $E_p$ for each month),

• the respective mean monthly grids of $E_r$ based on ASCE-standardized for short reference crop (Eq.1) from Aschonitis et al. (2017) (in the form of 12 raster datasets of $E_r$ for each month).





The validation procedure with the data of the 525 stations was performed by comparing the mean monthly and the mean annual benchmark values of $E_r$ (Eq.4) versus the original $E_p$ (Eq.1) and versus the corrected $E_{ps}$ Thornthwaite formula (Eq.11) taking into account the annual partial weighted average coefficients $C_{th}$ at the location of each station. The validation was made

separately for each database of stations (ECAD, AGBM, CIMIS) but also all together using the following five statistical criteria:

$$MAE = \frac{1}{N}\sum_{i=1}^{N}|S_i - O_i| \qquad (12)$$

$$ME = \frac{1}{N}\sum_{i=1}^{N}(S_i - O_i) \qquad (13)$$

$$RMSE = \sqrt{\frac{1}{N}\sum_{i=1}^{N}(S_i - O_i)^2} \qquad (14)$$

$$R_{Sqr} = \left[\frac{\sum_{i=1}^{N}(O_i - \overline{O_i})(S_i - \overline{S_i})}{\sqrt{\sum_{i=1}^{N}(O_i - \overline{O_i})^2 \sum_{i=1}^{N}(S_i - \overline{S_i})^2}}\right]^2 \qquad (15)$$

$$d = 1 - \frac{\sum_{i=1}^{N}(S_i - O_i)^2}{\sum_{i=1}^{N}(|S_i - \bar{O}_i| + |O_i - \bar{O}_i|)^2} \qquad (16)$$

where *MAE*: the mean absolute error, *ME*: the mean error, *RMSE*: the root mean square error, $R_{Sqr}$: the coefficient of determination and *d*: the index of agreement, *O*: the observed or benchmark value (i.e. $E_r$), *S*: the simulated value by the model (i.e. $E_p$ or $E_{ps}$), *N*: the number of observations, *i*: the subscript referred to each observation. The value of perfect fit is 0 for the

criteria *MAE*, *ME* and *RMSE* while is 1 for the criteria $R_{Sqr}$ and *d*. The values of *MAE*, *ME* and *RMSE* criteria have the same units with the observed and simulated data while $R_{Sqr}$ and *d* are unitless.

### 2.3 Evaluating the use of correction coefficients in aridity indices based on stations data

The role of the new corrected formula of Thornthwaite (Eq.11) as internal parameter of aridity indices was also evaluated against the original method (Eq.1). For this purpose, the $AI_{UNEP}$ (UNEP, 1997) and $AI_{TH}$ (Thornthwaite, 1948) aridity indices

were used. The difference between the two indices is that $AI_{UNEP}$ does not consider seasonality. The two indices estimated based on $E_r$ (Eq.4) were used as benchmark in order to compare the respective indices calculated with the original Thornthwaite $E_p$ (Eq.1) and the corrected $E_{ps}$ (Eq.11) using the 525 stations data. The evaluation was performed:

- by comparing the estimated aridity classes of 525 stations produced by the benchmark $AI_{UNEP}$ and $AI_{TH}$ values using $E_r$ versus the classes of the two indices using $E_p$ and $E_{ps}$, respectively.

- by comparing the respective values of the indices using 1:1 plots and the statistical metrics of Eqs.12-16.

   The $AI_{UNEP}$ aridity index is the simpler method for hydroclimatic analysis and it is given by the following equation:





$$AI_{UNEP} = \frac{P_y}{E_y} \qquad (17)$$

where $P_y$: mean annual precipitation (mm/year) and $E_y$: mean annual potential evapotranspiration (mm/year). The values of Eq.16 are classified according to the following (UNEP, 1997; Cherlet, 2018):

- $AI_{UNEP}$<0.05 → Hyper-arid
- 0.03≤$AI_{UNEP}$<0.2 → Arid
- 0.2≤$AI_{UNEP}$<0.5 → Semi-arid
- 0.5≤$AI_{UNEP}$<0.65 → Dry subhumid
- 0.65< $AI_{UNEP}$ → Humid

The classes for $AI_{UNEP}$>0.65 are usually given as one humid class. The UNEP index does not consider the effect of seasonal
variation of precipitation and potential evapotranspiration.

The $AI_{TH}$ aridity index is calculated as follows:

$$S = \sum_{i=1}^{12}(P_i - E_i) \quad and \quad D = \sum_{i=1}^{12}(E_i - P_i) \qquad (18a,b)$$

$$AI_{TH} = 100\frac{S - 0.6D}{E_y} \qquad (19)$$

where $P_i$ and $E_i$ are the monthly precipitation and potential evapotranspiration of month $i$, respectively. $S$ (mm y$^{-1}$) considers only the positive values of ($P_i$-$E_i$)>0, while ($P_i$-$E_i$)<0 are set 0. In the case of $D$ (mm y$^{-1}$), only the positive values of ($E_i$ -$P_i$)>0 are considered while for ($E_i$ -$P_i$)<0 are set 0. The various climatic types according to $AI_{TH}$ values are the following:

- -60>$AI_{TH}$ → Hyper-arid (HE)
- -60≤$AI_{TH}$<-40 → Arid (E)
- -40≤$AI_{TH}$<-20 → Semi-arid (D)
- -20≤$AI_{TH}$<0 → Dry sub-humid (C1)
- 0≤$AI_{TH}$<20 → Moist sub-humid (C2)
- 20≤$AI_{TH}$<40 → Low Humid (B1)
- 40≤$AI_{TH}$<60 → Moderate Humid (B2)
- 60≤$AI_{TH}$<80 → Highly Humid (B3)
- 80≤$AI_{TH}$<100 → Very Humid (B4)
- 100≤$AI_{TH}$ → Hyper-humid (A)





## 3. Results

### 3.1 Derivation and validation of the $C_{th}$ correction coefficients

The global map of the $C_{th}$ correction coefficient was developed following the procedure described in Section 2.2 and it is given in Fig.3. The validation of the derived $C_{th}$ coefficients was performed for each one of the three datasets of stations (California-CIMIS, Australia-AGBM, Europe-ECAD), separately, by comparing the performance of mean monthly values (Fig.S1a-f, supplementary material) and the performance of mean annual values (Fig.S2a-f, supplementary material) of $E_p$ (Eq.1) and $E_{ps}$ (Eq.11) versus the benchmark values of $E_r$ (Eq.4). The statistical criteria (Eqs.12-16) for both monthly and annual comparisons for each one of the three datasets of stations are given in Table 1. The respective monthly and annual comparisons after merging all the stations from the three datasets are also presented in Fig.6a-d. From the results shown in Figs.4,5,6 and Table1, it is observed a much better performance of $E_{ps}$ compared to the original Thornthwaite formula $E_p$ in all cases providing not only better monthly but also better annual reference evapotranspiration estimations that approximate the values of ASCE for short reference grass.

[FIGURE 3]

[TABLE 1]

[FIGURE 4]

### 3.2 Evaluating the use of $C_{th}$ coefficient in $AI_{UNEP}$ and $AI_{TH}$

The use of $C_{th}$ coefficients in $AI_{UNEP}$ and $AI_{TH}$ aridity indices was evaluated based on the raw data of all 525 stations (California-CIMIS, Australia-AGBM, Europe-ECAD).

The aridity classes of 525 stations given by the benchmark $AI_{UNEP}$ using $E_r$ were identical at 76% with the classes of the $AI_{UNEP}$ using $E_p$ and 93% identical with the classes of the $AI_{UNEP}$ using $E_{ps}$. Similarly, the aridity classes of 525 stations given by the benchmark $AI_{TH}$ using $E_r$ were identical at 52% with the classes of the $AI_{TH}$ using $E_p$ and 58% identical with the classes of the $AI_{TH}$ using $E_{ps}$. $E_{ps}$ showed better performance compared to $E_p$ at correctly identifying the aridity classes in both indices. The lower percentages of success in the case of $AI_{TH}$ for both $E_p$ and $E_{ps}$ are due to the double number of classes of $AI_{TH}$ in comparison to $AI_{UNEP}$. Merging the B and A classes of $AI_{TH}$ to one Humid class, as in the case of $AI_{UNEP}$, the successful identical codes are raised to 63% for $E_p$ and 76% for $E_{ps}$.

The 1:1 log-log plots of $AI_{UNEP}$ using $E_r$ versus the $AI_{UNEP}$ using $E_p$ and $E_{ps}$ are given in Fig.5a,b, respectively, while the same comparisons using $AI_{TH}$ are given in Fig.6a,b. The visual inspection of Figs.5,6 clearly shows that $E_{ps}$ outperforms the $E_p$ in the range of non-humid classes of both $AI_{UNEP}$ and $AI_{TH}$. In order to highlight this result, the statistical metrics (Eqs.12-16) were estimated after splitting the stations in two groups (non-humid and humid) based on the respective thresholds of humid classes of each index calculated using $E_r$ (Table 2). Table 2 verifies the better performance of $E_{ps}$ compared to $E_p$ in both $AI_{UNEP}$ and $AI_{TH}$ aridity indices for the non-humid classes.



On the other hand, the statistics showed that $E_p$ showed better performance in both $AI_{UNEP}$ and $AI_{TH}$ aridity indices for their respective humid classes. This result is of less importance since $E_{ps}$ showed better performance compared to $E_p$ at correctly identifying the aridity classes in both indices based on all stations despite the fact that the stations belonging to humid classes were more in both indices (Table 2). Moreover, in the case of $AI_{UNEP}$, there is only one Humid class ($AI_{UNEP} > 0.65$) and thus there is no point to compare the performance of $E_p$ and $E_{ps}$ from a statistical point of view since their values will always lead to the same classification code/characterization (i.e. Humid). In the case of $AI_{TH} > 20$, the same justification of $AI_{UNEP}$ could be used since the detailed division of five humid classes (B1, B2, B3, B4, A) provided by $AI_{TH}$ was proposed for the alternative use of the index as "humidity index" (Thornthwaite, 1948).

**[FIGURE 5]**

**[FIGURE 6]**

**[TABLE 2]**

## 4 Discussion

### 4.1 Validity of the derived $C_{th}$ for periods beyond the calibration period

The derivation of local $C_{th}$ coefficients at global scale was performed using the mean monthly grid datasets of 1950-2000 assuming stationary climate conditions, while the validation was performed using stations' raw data from California and Australia that are expanded up to 2016, and stations' raw data from Europe that are expanded up to 2020 (Table S1). The reasons for choosing the specific grid datasets for the derivation of $C_{th}$ coefficients are the following:

- They are in the form of high-resolution grids (30 arc-sec, ~1 km at equator), which have been developed using interpolation techniques that include the effects of latitude, longitude and elevation. These grids allow to derive more representative $C_{th}$ values for every position even when weather stations do not locally exist.
- They cover a large period of time (i.e. 1950-2000) so they can provide more representative mean annual p.w.a. $C_{th}$ values. The upper threshold of the year 2000 of these grids also allows the validation dataset of stations to be more valid since the larger part of their data is after 2000 and this reduces the possibility of having been used in grids' development.

On the other hand, several works have shown climate differences after 2000 (Hansen et al., 2010; McVicar et al., 2012a,b; Wild et al., 2013; Willet et al., 2014; Sun et al., 2017). Such changes could possibly affect the validity of $C_{th}$ coefficients and the final estimated values of $E_r$ for periods beyond 2000. For this reason, the $C_{th}$ values and the mean monthly $E_r$ values of the grids of Aschonitis et al. (2017) of the period 1950-2000 were extracted from the positions of all 525 stations and compared with the respective values of computed $E_r$ and $C_{th}$ using stations' raw data, which go beyond 2000. The results of this comparison are given in Figs.S3a,b (supplementary material) and clearly show that the gridded $E_r$ data and $C_{th}$ of 1950-2000 do not show serious deviations from their respective values for periods beyond 2000 allowing their safe use. Moreover, the fact that the original Thornthwaite (1948) formula was built before 1950 using data from the eastern and central USA and





that the $C_{th}$ values of the specific territories range between 0.9-1.1 for 1950-2000 (Fig.3), it is not only a verification of the $C_{th}$ derivation methodology but also an additional indication of a generalized temporal stability of $C_{th}$.

In the case of Fig.S3b, there is a distinctly deviated $C_{th}$ pair of values from the 1:1 line (point indicated by a red arow),
which is associated to a specific station belonging to the Centro de Investigación Atmosférica de Izaña. This station is an exceptional case since it is at the top of a mountain at 2371 m a.s.l. in Tenerife island. The derived $C_{th}$ of this station from the grid of the period 1950-2000 is almost half ($C_{th}$ value equal to 1.37) from the one estimated using stations' raw data ($C_{th}$ value equal to 2.44). This large difference is not the result of climate difference before and after 2000 but it is fully justified by the fact that the $C_{th}$ value of the grid corresponds to an area of ~1 km while the specific position of the station is at a very unique
position, which can be described as the most extreme position within this pixel. There are also 3 stations in Tenerife island at lowland areas where the derived $C_{th}$ values of 1950-2000 are in agreement with those estimated by the stations' raw data.

**4.2 Scale and other effects on the accuracy of the derived $C_{th}$**

The case of Izana station in Tenerife was the perfect example for triggering further investigation for the possible effects of
scale in similar environments with extremely variable topography. Investigating the individual stations with the larger % deviation of $E_{ps}$ from $E_r$, it was observed a relative systematic deviation in some stations of CIMIS-California database, which are concentrated in the coastline between Los Angeles and San Diego. The specific region is a narrow (~20-30 km) highly urbanized coastal zone of ~200 km, which is enclosed between the coastline and a hilly/mountainous zone. In the specific stations, the average of $C_{th}$ values of the period 1950-2000 from the position of these stations was 1.85, while the average of
$C_{th}$ values using their raw data was estimated at 1.46. Apart from the large topographic variation, another reason for the $C_{th}$ differences in these stations could be the bias that has been removed by clearing extreme flagged wind values in the data of CIMIS database, which are probably associated to hurricane or other extreme events in this region. The coastline region of California is strongly affected by hurricanes and the higher wind speeds during such events may have not been removed by the Sheffield et al. (2006) wind grids that were used by Aschonitis et al. (2017) to build the $E_r$ grids. This could justify the fact
that the gridded $C_{th}$ values of 1950-2000 at the positions of the stations are greater than the $C_{th}$ values estimated by their raw data from CIMIS after removing flagged extreme values.

An additional analysis based only on the stations of California was made to show that a wider regional mean value of $C_{th}$ coefficient could also be an additional option, especially when the whole territory is described by local $C_{th}$ coefficients that are only >1 or only <1 (in California all local $C_{th}$ coefficients are >1). For this analysis, the average value of $C_{th}$=1.66 was
estimated based on the values of local $C_{th}$ coefficients of 1950-2000 from the locations of all stations of CIMIS-California. The mean monthly and mean annual $E_{ps}$ values of these stations were computed using $C_{th}$=1.66 for all of them and compared with the respective $E_r$ values estimated with stations' raw data (Fig.S4a,b, supplementary material). The results of Fig.S4a,b showed that even a regional average of $C_{th}$ values for California can lead to better results of $E_{ps}$ compared to $E_p$ as it was given for monthly and annual estimations in Figs.S1a and S2a, respectively.




### 4.3 Justifications about the methodology for deriving annual $C_{th}$ correction coefficients based on partial weighted averages

The initial trials to derive annual correction coefficients $C_{th}$ of this study were made using the average value of the twelve monthly $c_{th,i}$ values of each $i$ month. This procedure led to unreasonably high values due to the extreme high values during

winter. An example of this problem based on the gridded data used in the calibration/derivation procedure, is given in Fig.S5a (supplementary material), which corresponds to a position close to Garda Lake in Italy (10.124° E, 45.45° N). According to Fig.S5a, the annual average of monthly $c_{th,i}$ values for this location is equal to 2.4 due to the extremely high values during winter and especially January. Using the 2.4 value as annual correction coefficient, the $E_{ps,i}$ value of July becomes equal to 338 mm, which is 203 mm larger from the respective $E_{r,i}$ value of July (Fig.S5a). The specific procedure for deriving annual

$C_{th}$ coefficients was rejected due to this problem. A second approach was to use the 12 pairs of monthly $E_r$ and $E_p$ for each position on the grid in order to perform regression analysis based on the form $y = a \cdot x$ without intercept based on the form of $E_r = C_{th} \cdot E_p$. An example of the specific procedure is given in Fig.S5b using the data of Fig.S5a, where the annual $C_{th}$ value was found equal to 0.98. The specific procedure provides annual $C_{th}$ values, which are always closer to the monthly coefficients of the warmer months since optimization algorithms try to minimize the total error, which is mainly originated by the months

that show larger evapotranspiration values. Despite the fact that the specific procedure pays less attention to the monthly $c_{th,i}$ values of colder months, it was considered acceptable since most of the hydroclimatic applications require higher accuracy to the larger evapotranspiration values rather to the lower ones.

A similar approach with the one of Fig.S5b was performed by Cristea et al. (2013) for deriving annual correction coefficients for the Priestley-Taylor method for 106 stations across the contiguous USA. The correction coefficients were

estimated for each station by minimizing the sum of the squared residuals between Priestley-Taylor and the benchmark FAO-56 considering data only for the period April-September (warmer semester). The obtained optimized values of the correction coefficients for each station were then interpolated to produce a map of the Priestley-Taylor correction coefficients. For our study, the specific procedure was found to be extremely demanding in computing requirements since it was impossible to be performed pixel by pixel (777.6 million pixels) with a conventional computer unit for the whole globe using as input 24 rasters

of extremely high resolution (~1 km) with total size of ~70GB. In order to solve this problem, the method of partial weighted average (Eqs.5-10) developed by Aschonitis et al. (2017) was used, which provides similar results to the regression analysis of $y = a \cdot x$ but allows to perform calculations step by step with a conventional computer unit in GIS environment using large gridded databases. For the data of Fig.S5a, the partial weighted average method provided a $C_{th}$ value equal to 0.99, which is almost equal to 0.98 of Fig.S5b. The method of partial weighted average is also extremely efficient since it is not restricted

only to the warmer semester or to any other predefined period like the case of Cristea et al. (2013) since it controls all months one by one using the threshold of 45 mm month$^{-1}$, which is more appropriate for global applications and especially for applications of high-resolution data, giving the appropriate weight to the months with significant values of evapotranspiration.

The threshold of 45 mm month$^{-1}$ was derived empirically after analysing many datasets using monthly and mean monthly data. In the case of monthly data, a representative example is given in Figs.S6a,b (supplementary material) using the



monthly data of Embrun station in France (6.50º E, 44.57º N) 1980-2020. Fig.S6a shows the box-whisker plots of monthly $E_{r,i}$
values of the station, while Fig.S6b shows the respective box-whisker plots of monthly $c_{th,i}$ values. The maximum $c_{th,i}$ values
of December, January and February are outside the plot of Fig.S6b with values 30.1, 129.4 and 210.1, respectively. Figs.S6a,b
show that the monthly $c_{th,i}$ values of months with $E_{r,i} < 45$ mm month$^{-1}$ are extremely unstable and their mean monthly value,
even if it seems normal, cannot guarantee its safe use.  In the case of mean monthly data, a representative example is given in

Figs.7, where the 6300 mean monthly $c_{th,i}$ values derived by the raw data of the 525 stations were plotted against their respective
mean monthly $E_r$ values using a 2D density scatter plot. Fig.7 shows that the mean monthly $c_{th,i}$ values of the stations start to
exhibit extremely high dispersion below the threshold of 45 mm month$^{-1}$ with values reaching one order of magnitude larger
than unity. In the case where there is a location where all months show $E_r$ or $E_p$ values below 45 mm month$^{-1}$, it is suggested
either to use the non-zero $C_{th}$ value of the closer location in the map of Fig.3 or to use directly the original Thornthwaite

formula without correction.

**[FIGURE 7]**

## 4 Data availability

The produced global database of local $C_{th}$ coefficients of this study has been archived in PANGAEA and can be assessed using
the following link: https://doi.pangaea.de/10.1594/PANGAEA.932638 (Aschonitis et al., 2021). The database is provided at 5

different resolutions (30 arc-sec, 2.5 arc-min, 5 arc-min, 10 arc-min, 0.5 deg). The coarser resolutions are provided in order to
cover the observed resolution range in the initial climatic data used for developing the published $E_r$ gridded data by Aschonitis
et al. (2017) (e.g., the temperature data of Hijmans et al. (2005) were provided at 30 arc-sec resolution, while the solar radiation,
humidity and wind speed data of Sheffield et al. (2006) were provided at 0.5 deg resolution and rescaled to 30 arc-sec using
bilinear interpolation). The data of different resolutions can be used as a tool to assess uncertainties associated to temperature

variation effects within different resolution pixels or to estimate average values of the coefficients for larger territories, which
have problems at coarse resolutions (e.g., coastlines or islands that do not exist in 0.5 degree resolution) taking into account
the concept and concerns of Daly et al. (2006).

## 5 Conclusions

A global database of local correction coefficients for improving the performance of the  monthly temperature-based

Thornthwaite potential evapotranspiration method was built using gridded data covering the period 1950-2000. The method
for developing the correction coefficients was based on partial weighted averages of their respective mean monthly values
estimated as the monthly ratios between the benchmark ASCE-standardized $E_r$ method (former FAO-56) versus the original
Thornthwaite $E_p$. The correction coefficients were produced as partial weighted averages of monthly $E_r/E_p$ ratios by setting the
ratios' weight according to the monthly $E_r$ magnitude and by excluding colder months because the $E_r/E_p$ ratio becomes highly



unstable for low temperatures. The correction coefficients were validated using raw data from 525 stations of California,
       Australia and Europe that include independent data beyond 2000 up to 2020. The results showed that the corrections
       coefficients significantly improved the monthly and annual results of original Thornthwaite method $E_p$. The use of $E_p$ with or
       without correction coefficients was also evaluated through their use in the aridity indices of Thornthwaite and UNEP versus
       the respective indices estimated based on the benchmark ASCE-standardized $E_r$. The results showed again that the correction

coefficients significantly improved the performance of the indices compared to the original Thornthwaite method especially
       in non-humid environments. The global database of local correction coefficients supports applications of reference
       evapotranspiration and aridity indices assessment with minimum data requirements (i.e. mean temperature) for locations where
       climate data are limited. Uncertainties in the values of correction coefficients were observed in regions of high topographic
       variability and a recommendation for such cases is the use of a regional average of correction coefficients or the use of local

$C_{th}$ values based on the available coarser resolutions provided in the database. The methods and results presented in this study
       and the observed uncertainties can be used as a base for future works focusing on: (a) the validation of the correction
       coefficients for other places in the world, (b) comparison with other models of low data requirements, (c) use of the p.w.a.
       method for recalibrating correction coefficients using station or climate models' data of recent periods.

**Supplementary material.** Supplementary information related to the article is given in the following supplementary file (to be
       added by the journal).

       **Author contributions.** The idea behind the work was conceived by VA, the data processing was made by V.A and DT while
       quality control, visualization and writing were completed by VA, DT, MCG, MCV.


       **Competing interests.** The authors declare that they have no conflicts of interest.

       **Financial support.** This research has not received funding.

**Review statement.** This paper was edited by … and reviewed by …. referees.

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

**Tables**

**Table 1.** Statistical metrics (Eqs.12-16) for the comparisons between $E_p$ vs. $E_r$ and $E_{ps}$ vs. $E_r$ for CIMIS-California, AGBM-Australia and ECAD-Europe stations (the unit for *MAE*, *ME*, *RMSE* is mm month$^{-1}$).

| | California | | Australia | | Europe | |
|---|---|---|---|---|---|---|
| | $E_p$ vs. $E_r$ | $E_{ps}$ vs. $E_r$ | $E_p$ vs. $E_r$ | $E_{ps}$ vs. $E_r$ | $E_p$ vs. $E_r$ | $E_{ps}$ vs. $E_r$ |
| Metrics based on mean monthly values | | | | | | |
| No. records | 720 | 720 | 960 | 960 | 4620 | 4620 |
| *MAE* | 40.3 | 22.6 | 64.6 | 45.2 | 14.5 | 11.9 |
| *ME* | -39.7 | 4.1 | -60.5 | 17.3 | -7.4 | -6.9 |
| *RMSE* | 46.4 | 31.0 | 74.2 | 63.7 | 20.1 | 15.3 |
| $R_{Sqr}$ | 0.852 | 0.858 | 0.624 | 0.746 | 0.824 | 0.919 |
| $d$ | 0.847 | 0.948 | 0.743 | 0.867 | 0.945 | 0.972 |
| Metrics based on mean annual values | | | | | | |
| No. records | 60 | 60 | 80 | 80 | 385 | 385 |
| *MAE* | 476.2 | 142.1 | 730.5 | 256.8 | 116.6 | 101.6 |
| *ME* | -476.2 | 49.8 | -726.5 | 208.0 | -89.3 | -83.1 |
| *RMSE* | 500.1 | 177.9 | 800.2 | 317.0 | 184.7 | 126.0 |
| $R_{Sqr}$ | 0.717 | 0.603 | 0.526 | 0.812 | 0.785 | 0.879 |
| $d$ | 0.501 | 0.845 | 0.571 | 0.906 | 0.728 | 0.94 |









**Table 2.** Statistical metrics (Eqs.12-16) for the comparisons between $E_p$ vs. $E_r$ and $E_{ps}$ vs. $E_r$ when they are applied in the (a) $AI_{UNEP}$ and (b) $AI_{TH}$ aridity indices by dividing the 525 stations to two groups based on non-humid or humid classes of each index (*MAE*, *ME*, *RMSE* are unitless as the indices).

| | $E_p$ vs. $E_r$ | $E_{ps}$ vs. $E_r$ | $E_p$ vs. $E_r$ | $E_{ps}$ vs. $E_r$ |
|---|---|---|---|---|
| (a) | *Stations with $AI_{UNEP}\leq$0.65\* (non-humid)* | | *Stations with $AI_{UNEP}>$0.65\* (humid)* | |
| No. stations | 197 | 197 | 328 | 328 |
| *MAE* | 0.169 | 0.036 | 0.151 | 0.264 |
| *ME* | 0.169 | 0.003 | 0.035 | 0.233 |
| *RMSE* | 0.194 | 0.056 | 0.264 | 0.376 |
| $R_{Sqr}$ | 0.867 | 0.893 | 0.875 | 0.932 |
| *d* | 0.773 | 0.969 | 0.963 | 0.950 |
| (b) | *Stations with $AI_{TH}\leq$20\* (non-humid)* | | *Stations with $AI_{TH}>$20\* (humid)* | |
| No. stations | 257 | 257 | 268 | 268 |
| *MAE* | 12.8 | 6.3 | 14.9 | 26.7 |
| *ME* | 12.7 | 3.6 | 3.0 | 24.2 |
| *RMSE* | 15.1 | 10.0 | 26.6 | 39.4 |
| $R_{Sqr}$ | 0.842 | 0.882 | 0.872 | 0.928 |
| *d* | 0.855 | 0.939 | 0.962 | 0.945 |

\*estimated by the benchmark $E_r$.





**FIGURES**


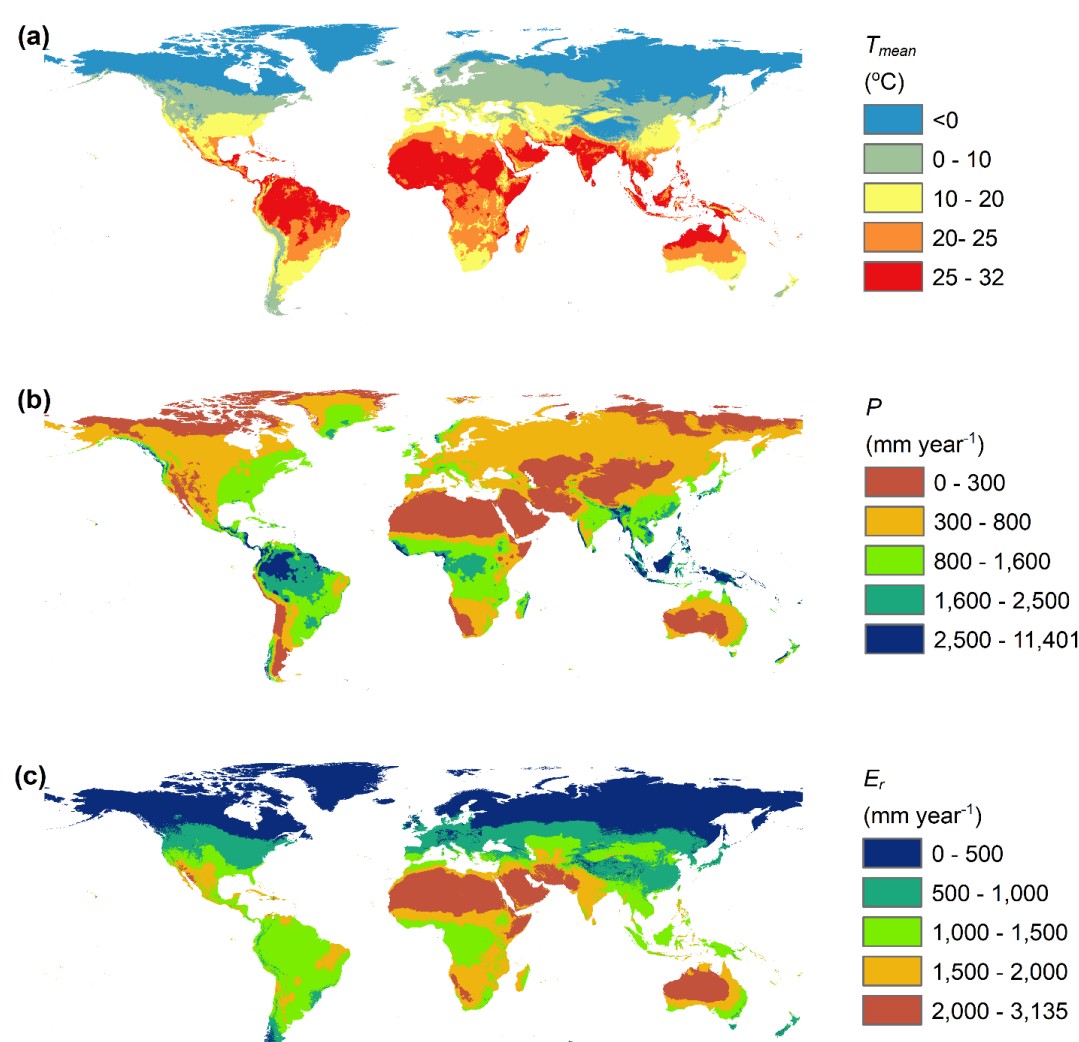

**Figure 1. (a)** Mean annual temperature for the period (Hijmans et al., 2005), **(b)** Mean annual precipitation for the period (Hijmans et al., 2005), **(c)** mean annual reference evapotranspiration of ASCE-standardized method for short reference crop for the period (Aschonitis et al., 2017) of 1950-2000.



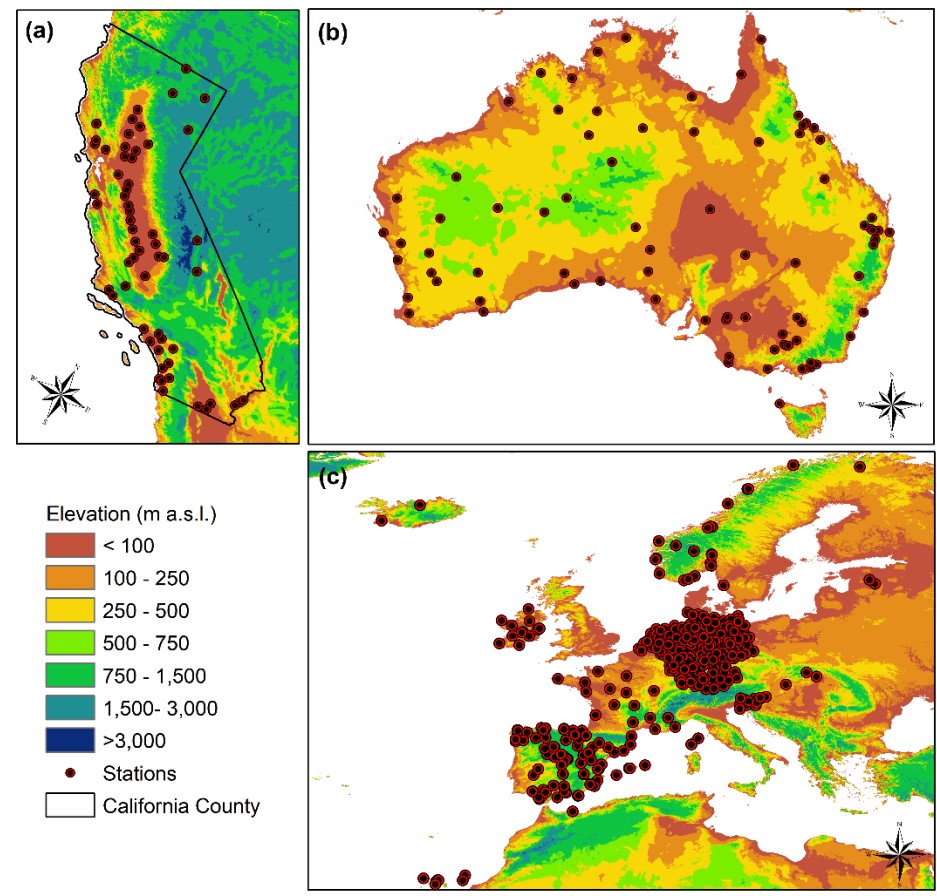


**Figure 2. (a)** 60 stations of California from CIMIS database, **(b)** 80 stations of Australia from AGBM database, and **(c)** 385 stations of Europe from ECAD database.

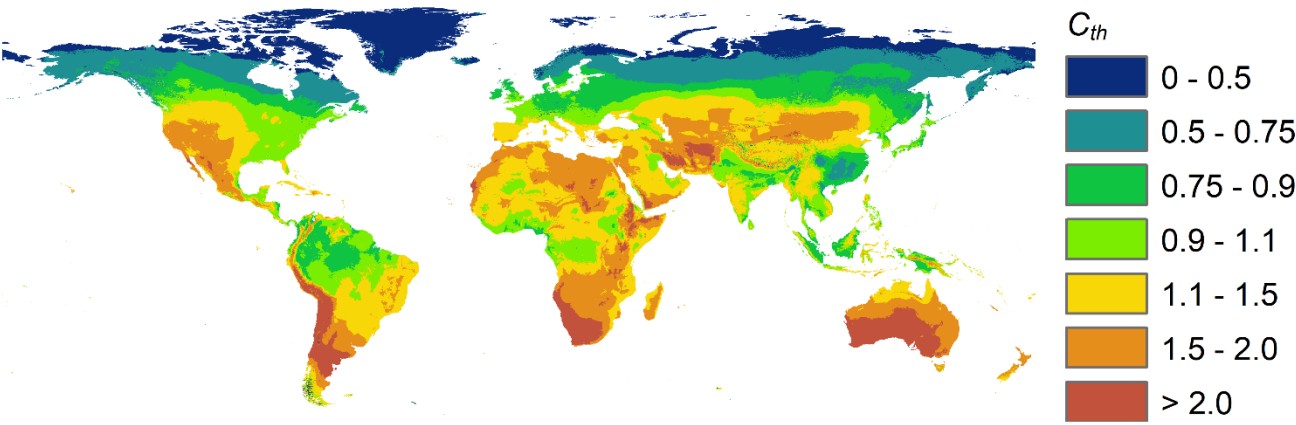

**Figure 3.** Global map of the annual partial weighted average $C_{th}$ coefficients.



**Figure 4. (a)** 1:1 plots of mean monthly $E_p$ versus mean monthly $E_r$, **(b)** mean monthly $E_{ps}$ versus mean monthly $E_r$, **(c)** mean annual $E_p$ versus annual monthly $E_r$, **(d)** mean annual $E_{ps}$ versus annual monthly $E_r$, using the data of all 525 stations from the three databases of CIMIS, AGBM, ECAD.



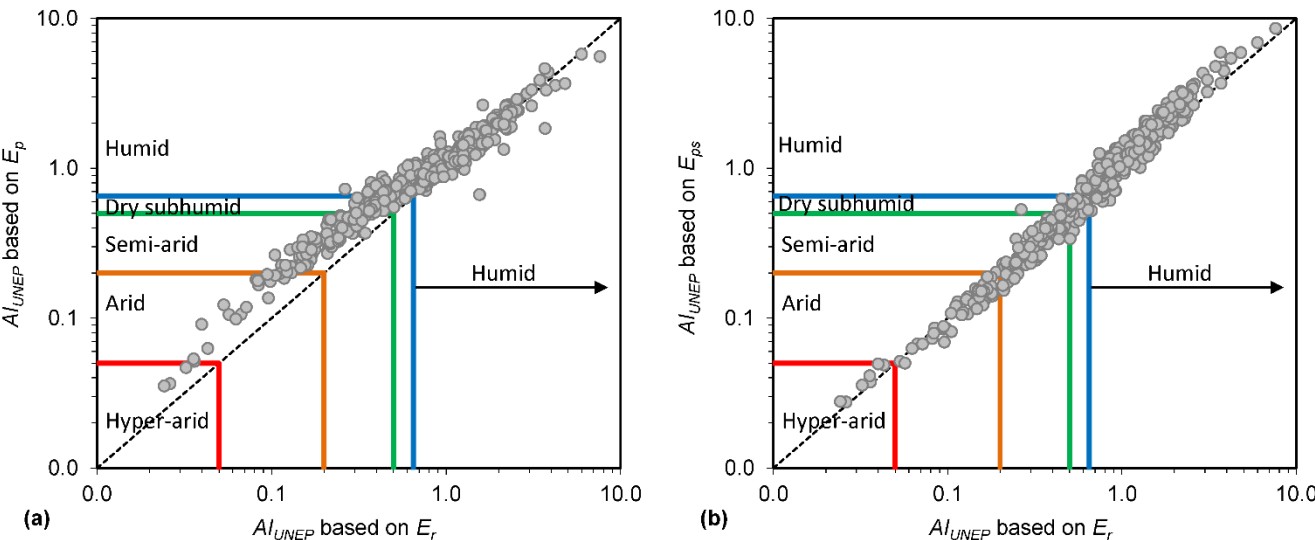

**Figure 5. (a)** 1:1 log-log plots of $AI_{UNEP}$ using mean monthly $E_p$ versus $AI_{UNEP}$ using mean monthly $E_r$, **(b)** $AI_{UNEP}$ using mean monthly $E_{ps}$ versus mean monthly $E_r$ using the data of all 525 stations from the three databases of CIMIS, AGBM, ECAD.


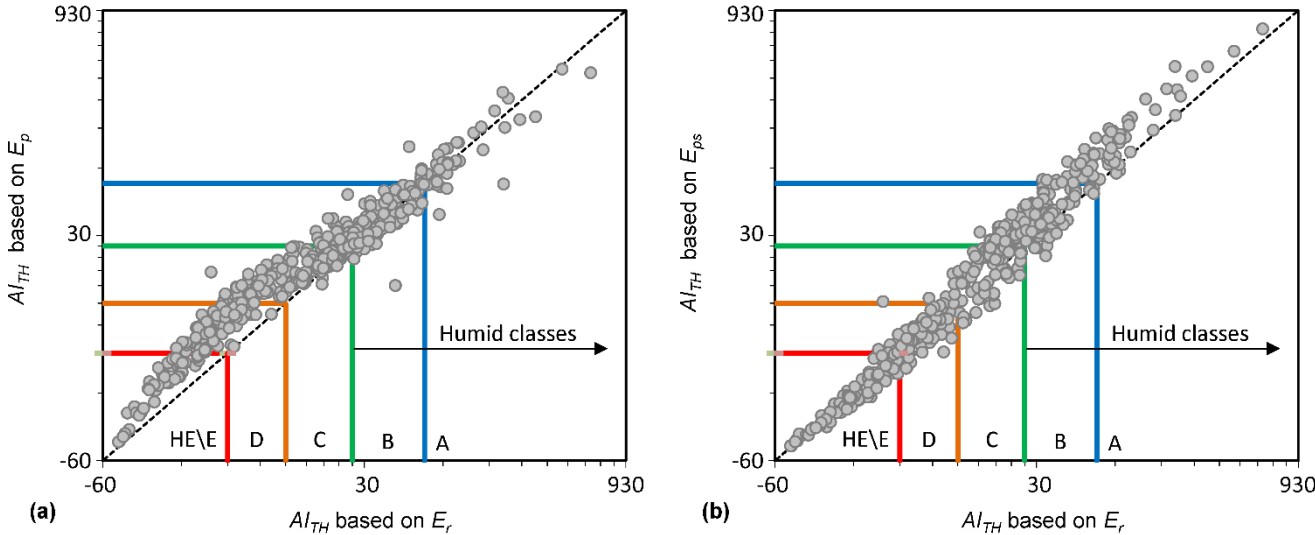

**Figure 6. (a)** 1:1 log-log plots of $AI_{TH}$ using mean monthly $E_p$ versus $AI_{TH}$ using mean monthly $E_r$, **(b)** $AI_{TH}$ using mean monthly $E_{ps}$ versus mean monthly $AI_{TH}$ using the data of all 525 stations from the three databases of CIMIS, AGBM, ECAD.




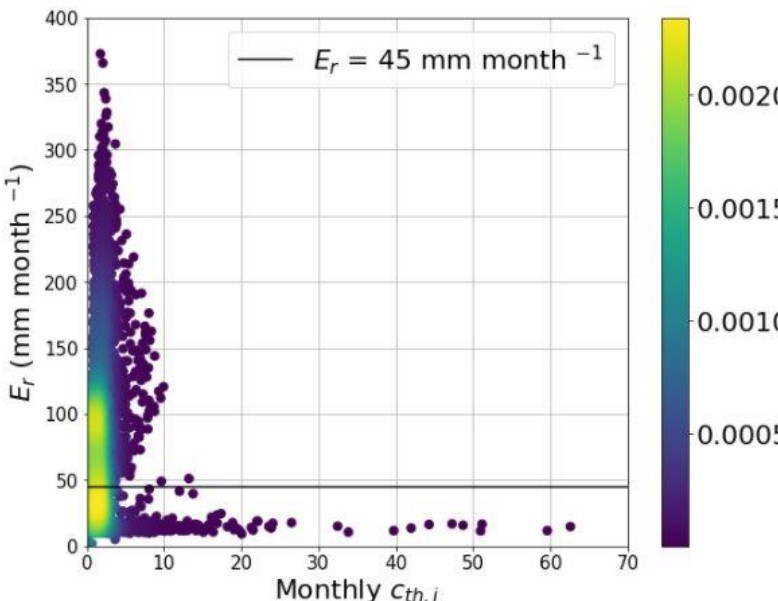

**Figure 7.** 2D scatter density plot between the 6300 mean monthly $c_{th,i}$ values versus the respective mean monthly $E_r$ values derived by the raw data of the 525 stations ($c_{th,i}$ =0 or non-defined due to $E_r$ and/or $E_p$ =0 were not included in the graph).
