# Peer review of "Correcting Thornthwaite potential evapotranspiration using a global grid of local coefficients to support temperature-based estimations of reference evapotranspiration and aridity indices"

_Earth System Science Data, 2021_

## Author Response (AR3)

**Correcting Thornthwaite potential evapotranspiration using a global grid of local coefficients to support temperature-based estimations of reference evapotranspiration and aridity indices**

Vassilis Aschonitis[1], Dimos Touloumidis[2], Marie-Claire ten Veldhuis[2], Miriam Coenders-Gerrits[2]

[1]Soil and Water Resources Institute, Hellenic Agricultural Organization – DEMETER, Thessaloniki – Thermi, 57001, Greece
[2]Water Resources Section, Delft University of Technology, Stevinweg 1, 2628 CN Delft, the Netherlands

*Correspondence to*: Vassilis Aschonitis (v.aschonitis@swri.gr)

**COMMENTS**

RC1 Anonymous Referee #1

The paper is well-written, and the methods, analyses and results are clearly presented. The results can be used in solving the practical problems. This reviewer only suggests inclusion of few relevant publications. The list is given below.

Bautista F, Bautista D, Delgado-Carranza C (2009) Calibration of the equations of Hargreaves and Thornthwaite to estimate the potential evapotranspiration in semi–arid and subhumid tropical climates for regional applications. Atmosfera 22(4):331–348

https://doi.org/10.1007/s00704-019-02873-1

https://doi.org/10.1016/j.atmosres.2021.105727

**Response:**

We thank the reviewer for the positive evaluation. The three citations suggested by the reviewer were incorporated in the revised manuscript.

1. Bautista F, Bautista D, Delgado-Carranza C (2009) Calibration of the equations of Hargreaves and Thornthwaite to estimate the potential evapotranspiration in semi–arid and subhumid tropical climates for regional applications. Atmosfera 22(4):331–348
2. Trajkovic, S., Gocic, M., Pongracz, R., Bartholy, J.. Adjustment of Thornthwaite equation for estimating evapotranspiration in Vojvodina. Theor Appl Climatol 138, 1231–1240 (2019). https://doi.org/10.1007/s00704-019-02873-1
3. Nikolaos D. Proutsos, Ioannis X. Tsiros, Panagiotis Nastos, Alexandros Tsaousidis, 2021. A note on some uncertainties associated with Thornthwaite's aridity index introduced by using different potential evapotranspiration

methods, Atmospheric Research, Volume 260, 105727, https://doi.org/10.1016/j.atmosres.2021.105727.

CC1 Sun Shijun

The paper is well-written, and the methods, analyses and results are clearly presented. However, the aim of this study is to correct the formula of monthly Thornthwaite potential evapotranspiration (Ep), maybe the daily reference evapotranspiration more significant. Many many studies had been conducted to explore temperature based reference evapotranspiration model, including classical formula, machine learning model, and deep learning model. In present, the weather variables is more easier obtained than before. Under this condition, maybe correcting monthly potential evapotranspiration model had its significant, but I do not think it is big.

**Response:**

We respect the opinion of the reviewer, but we disagree with his comment based on the following three aspects:

1. **Regarding the monthly or daily time step.** The paper focuses on the correction of a specific methodology: the original Thornthwaite potential evapotranspiration formula; which is a formula that inherently provides only monthly estimations and not daily. Thus, daily reference evapotranspiration comparisons could not be anyway performed based on the corrections of this formula against FAO Penman-Monteith. The paper also deals with aridity indices that use monthly records and not daily. The potential evapotranspiration method of Thornthwaite is still used in these indices and the corrected version of it can significantly improve their performance using only temperature data (as it was shown by the results of this study) reducing the differences that are observed in aridity indices using different methods as indicated by Proutsos et al. (2021).

2. **Regarding the machine learning models.** We provided a model that uses one independent variable with correction coefficients for each position in the globe. Machine learning methods are useful for describing phenomena with many independent variables and not just one (i.e., mean temperature monthly or even daily). If we assume that this was not a problem, a machine learning method should be calibrated-validated based on the whole global grid of records providing a unique global model. This would require the machine learning model to describe internally the different combined local effects of the non-participating parameters (wind speed, relative humidity, solar radiation) for each position in the globe, which in our opinion would be quite difficult (if not impossible). In this case, the global machine learning model would make an average description of the different combined local effects of the non-participating parameters (wind speed, relative humidity, solar radiation) for the whole globe that would lead to large errors in the much colder and much warmer regions. We know this, because we tried it during preliminary trials (non-published) for a previous paper (https://doi.org/10.5194/essd-9-615-2017) where machine learning techniques were

not able to reach the accuracy of Hargreaves-Samani method corrected with local coefficients. That is why we developed the methodology of local weighted coefficients in that paper and used it also in this paper. Other option would be to build one machine learning model for each pixel (or cluster of pixels with similar conditions) covering the whole globe using only mean temperature (again the problem of one independent variable and quite difficult as task even with more independent variables for a global application). Other option that we tested using machine learning techniques was to describe the local $c_{th}$ factors using latitude, longitude and altitude, but the results were not as good as the direct use of the local coefficients.

3. **Regarding the fact that the weather variables are nowadays easier accessible.** We agree that every year the number of stations that provide complete datasets of climate variables is increasing. However, the spatial coverage over the globe is still not sufficient (e.g., in developing countries the number of stations is limited). Moreover, even in developed countries, the current number of stations is not sufficient to capture the climatic variation within watersheds. For example, the number of meteorological stations in mountainous areas* is too limited. Lastly, even in many developed countries complete datasets are also not freely or easily available.

* We have to stress the fact that the initial databases of Hijmans et al. for temperature and precipitation at 1 km resolution, which were used in this study, were built using ANUSPLIN-SPLINA package for interpolation, using latitude, longitude, and elevation as independent variables. This allowed to describe better temperature and precipitation patterns in mountainous areas.

Additional corrections-improvements in grammatical/typo errors were also made at the following lines: 16, 68-69, 74-76, 83-89, 98, 117, 121, 132, 166, 174-175, 223, 230, 239, 242, 281, 284-286, 293, 319, 338 plus three new references in the reference list (lines 425, 490, 515).

**The additional correction requested by the editor at lines 295 and beyond regarding hurricanes was considered. The text was corrected according to the suggestions.**